# A Training Program Using Modified Joystick-Operated Ride-on Toys to Complement Conventional Upper Extremity Rehabilitation in Children with Cerebral Palsy: Results from a Pilot Study

**DOI:** 10.3390/bioengineering11040304

**Published:** 2024-03-23

**Authors:** Sudha Srinivasan, Patrick Kumavor, Kristin Morgan

**Affiliations:** 1Physical Therapy Program, Department of Kinesiology, University of Connecticut, Storrs, CT 06269, USA; 2Institute for Collaboration on Health, Intervention, and Policy (InCHIP), University of Connecticut, Storrs, CT 06268, USA; 3The Institute for the Brain and Cognitive Sciences (IBACS), University of Connecticut, Storrs, CT 06268, USA; 4Biomedical Engineering Department, University of Connecticut, Storrs, CT 06268, USA; patrick.d.kumavor@uconn.edu (P.K.); kristin.2.morgan@uconn.edu (K.M.)

**Keywords:** children, hemiplegia, joystick-operated ride-on toys, cerebral palsy, rehabilitation, innovative interventions, technology-based aids, upper extremity, motor function, accelerometry

## Abstract

The pilot study assessed the utility of a training program using modified, commercially available dual-joystick-operated ride-on toys to promote unimanual and bimanual upper extremity (UE) function in children with cerebral palsy (CP). The ride-on-toy training was integrated within a 3-week, intensive, task-oriented training camp for children with CP. Eleven children with hemiplegia between 4 and 10 years received the ride-on-toy training program 20–30 min/day, 5 days/week for 3 weeks. Unimanual motor function was assessed using the Quality of Upper Extremity Skills Test (QUEST) before and after the camp. During ride-on-toy training sessions, children wore activity monitors on both wrists to assess the duration and intensity of bimanual UE activity. Video data from early and late sessions were coded for bimanual UE use, independent navigation, and movement bouts. Children improved their total and subscale QUEST scores from pretest to post-test while increasing moderate activity in their affected UE from early to late sessions, demonstrating more equal use of both UEs across sessions. There were no significant changes in the rates of movement bouts from early to late sessions. We can conclude that joystick-operated ride-on toys function as child-friendly, intrinsically rewarding tools that can complement conventional therapy and promote bimanual motor functions in children with CP.

## 1. Introduction

Bimanual coordination refers to the spatiotemporal coupling of movements of both upper extremities (UEs) for completion of goal-directed activities [1]. Bimanual tasks may involve symmetrical or asymmetrical movements of both arms [2]. Regardless of specific task constraints, both UEs work synergistically as a single functional unit through precise coordination at the neural, neuromuscular, and biomechanical levels [3]. A large proportion of activities of daily living related to self-care, work/academics, and leisure (e.g., tying shoelaces, buttoning a shirt, opening a jar, using cutlery, cutting a piece of paper using scissors, stringing beads, lifting a heavy tray, etc.) require bimanual coordination [4]. In typically developing children, bimanual coordination begins to develop in infancy and continues to be refined over early childhood [5,6,7]. Over the course of development, children learn to differentiate the use of their dominant and non-dominant UEs (i.e., as a stabilizer vs. mobilizer) during bimanual activities, which is termed as role-differentiated bimanual manipulation [2,8]. This division of labor affords the efficient and timely completion of a multitude of everyday bimanual tasks [1].

In contrast to typically developing children, children with unilateral Cerebral Palsy (CP) or hemiplegia struggle with bimanual coordination [9,10]. These difficulties are attributed to multiple factors, including (a) impaired sensorimotor control, muscle weakness, and spasticity on one side of the body; (b) poor sensory function on the affected side that impairs anticipatory and reactive control of bimanual actions; (c) attentional, motor planning, and motor learning-related deficits; and (d) damage to neural substrates that underlie the spatiotemporal coupling of movements of both UEs during bimanual activities, including the supplementary motor cortex, parietal lobe, and corpus callosum [4,11,12,13,14,15,16,17]. Impaired bimanual coordination skills in children with hemiplegia ultimately contribute to their functional limitations in participating successfully across home, school, and other community settings [18]. 

Within the realm of pediatric rehabilitation, two prominent evidence-based approaches (namely modified Constraint Induced Movement Therapy (mCIMT) and Bimanual Training (BT)) have been tested for their effects on bimanual skills in children with hemiplegia [17,19,20,21]. By forcing the child to use their affected/non-dominant UE through constraint of their unaffected/dominant UE, mCIMT can help children overcome the “developmental disuse” of their affected arm and increase awareness of that UE [21]. Improvements in unimanual capacity and “priming” of the affected UE through intensive goal-directed practice during mCIMT have been proposed to likely carry over to greater spontaneous use of the affected UE during bimanual tasks [21,22]. On the other hand, paradigms involving BT promote the practice of bimanual tasks that closely mimic everyday functional activities [17,23]. Recently, it has been suggested that mCIMT and BT may likely have “synergistic” effects on UE function [22]. Therefore, hybrid paradigms that combine mCIMT and BT in different schedules have been implemented to promote UE function in children with hemiplegia [22,24,25,26,27,28]. The current study was conducted within such a hybrid 3-week summer camp for children with CP. 

Current evidence on UE rehabilitation approaches in CP suggests that high dose paradigms that foster intensive, task-oriented practice for several hours every day for multiple weeks produce meaningful improvements in function [23,29,30]; however, it is challenging to sustain child engagement and motivation during such intensive paradigms [31,32,33,34]. Over the last several years, our research group has been exploring the use of child-friendly, easy-to-implement, technology-based adjuncts, specifically joystick-operated ride-on toys, to incentivize UE use, boost treatment dosing, and promote task-oriented UE practice through incrementally challenging navigational games [35,36,37]. Previously, we reported that a training program using modified, single joystick-operated ride-on toys (with controls provided on the child’s affected side) was well-received by children, caregivers, and clinicians. The training was feasible for implementation within an intensive mCIMT program, and in combination with mCIMT, it led to improvements in affected UE use and motor functions assessed using standardized motor tests, arm accelerometry, and video-based measures [36,37]. However, given the importance of bimanual coordination for activities in daily living, the present study will explore the effects of an innovative and playful bimanual ride-on-toy training program among children with CP. The ride-on toys used in this study are controlled using two joysticks, requiring children to use both arms together in a synchronized fashion to drive the toy. The Dual-Joystick-operated ride-on-toy navigation Training (DJT) program tested in this pilot study was provided as one of the daily training activities at a 3-week hybrid (based on the principles of mCIMT and BT) training camp for children with CP. The manualized DJT protocol involved structured navigational tasks and UE challenges, both of which promoted bimanual coordination skills. We report on the combined effects of DJT and the hybrid programming (mCIMT + BT) on standardized, objective, and video-based measures of unimanual and bimanual UE motor function. We hypothesized that, by following the comprehensive camp-based training inclusive of DJT, children will increase the independent use of their affected UE both within and outside the DJT sessions, demonstrating improvements in unimanual and bimanual motor capacity. 

## 2. Materials and Methods

### 2.1. Study Design and Participants 

The single group quasi-experimental pretest−post-test study was conducted at the Lefty and Righty camp of Connecticut (LARC), held within the premises of a local middle school, using a convenience sampling approach. The camp director sent out informational fliers about the study to families who enrolled in the camp. We conducted a screening interview with interested families to assess their child’s eligibility for participation in the study. We included children with CP with clear asymmetry in upper extremity function, and we excluded children who exceeded the weight limits of the ride-on toys (150 lbs.), who were unable to sustain supported sitting for 20 min, who had a history of UE surgery/Botox/injury in the last 6 months, and those with uncorrected, profound vision impairments. 

Eleven children with hemiplegia between 4 and 10 years (4 males, 7 females; 6 children with right hemiparesis and 5 children with left hemiparesis; Age in years—Mean (SE): 6.46(0.64)) participated in the study. In terms of racial-ethnic distribution, 8 families were Caucasian Non-Hispanic, 1 family was Caucasian−Hispanic, and 2 families were multi-racial (one family mixed White and Asian, and one family mixed Korean, Puerto-Rican, Irish, and Polish). Study participants demonstrated a mild to moderate degree of impairment in their ability to use their hands for daily activities (Manual Ability Classification System (MACS) scores: 2 children at MACS Level I, 2 children at MACS Level II, and 7 children at MACS Level III). The MACS classifies children’s use of their hands to handle objects during daily activities on a 5-point scale, with lower scores indicating better abilities [38]. 

The study was approved by the Institutional Review Board at the University of Connecticut (IRB Protocol #: H21-0019). Parents signed off on the parental permission form prior to the start of the study. All children participating in the study provided assent prior to the start of the study, as well as before every testing and training session. 

### 2.2. Procedures 

#### 2.2.1. Camp Structure 

The LARC is an annual 3-week, intensive summer camp based on principles of mCIMT and BT that focuses on using playful gross and fine motor activities to promote UE motor functions among children with hemiplegia. The camp training was provided over a 3-week duration, 5 days every week, 6 h/day. Camp staff included physical therapists, occupational therapists, and trained paraprofessional aides. Each child worked one-on-one with a staff member over the course of the camp. The camp activities were aligned with child-friendly themes, such as children’s animated movies, popular sports, and common professions that children aspire to pursue in the future. Despite a structured schedule designed for every camp day, training activities were optimally tailored by clinicians to suit the needs of each individual child. The camp followed a hybrid structure where children spent roughly 85% of the camp day with their unaffected UE constrained within a removable thermoplastic cast. During this time, children were encouraged to use their affected UE for gross motor (e.g., weight bearing, mat activities, ball play, etc.) and fine motor activities (e.g., building games, art-craft activities, Play-Doh, etc.). During the remaining 15% of camp time, children removed their cast on the unaffected side and were encouraged to engage in bimanual activities that required simultaneous use of both UEs, with an emphasis on training functional daily tasks (e.g., feeding, toileting, dressing, etc.). During this phase, children practiced symmetrical and asymmetrical tasks that required using their affected UE as a stabilizer and a mobilizer. The dual-joystick-based ride-on-toy navigation program was incorporated into the bimanual training protocol at the camp. 

#### 2.2.2. Dual-Joystick-Based Ride-On-Toy Navigation Training (DJT)

Each child received DJT for 20–30 min every day at camp. We used commercially available ride-on toys, namely, the Wild Thing (Fisher-Price, East Aurora, New York, USA) and the Huffy Green Machine Vortex (Huffy, Dayton, OH, USA), for the training sessions (see Figure 1). The toys were modified to increase postural support (i.e., addition of forearm and leg support plates for the Huffy and external frame of PVC pipes for the Wild Thing), improve access to joysticks (i.e., raising the joysticks and providing foam balls if needed on top of cylindrical joysticks to improve grasp of the joysticks), and modulate toy speed (i.e., the toy control systems were altered to provide options to drive the toy at 3 speeds: slow, medium, and fast). 

The DJT sessions were aligned with camp themes and structured to include 2 components: navigation (~50% of session time) and goal-directed gross and fine motor UE tasks (~50% of session time). Please note that the DJT protocol was developed through previous pilot work in this area, with feedback being obtained from stakeholders (i.e., children, clinicians, and caregivers) during our pilot studies [36,37]. During navigation, children were encouraged to use both UEs simultaneously to maneuver the toy and navigate through courses/paths within their physical environment (i.e., moving forward, backward, turning right and left, navigating obstacle courses and paths of varying shapes such as straight, circular, and slalom paths). During goal-directed tasks, children engaged in progressive UE challenges that trained reach, grasp, manipulation, and release skills on the affected side. Specifically, tasks were designed to encourage children to use their affected UEs in the role of a stabilizer and a mobilizer during bimanual tasks (see Figure 2). 

Tasks were progressed to provide children with a “just right” challenge based on the principles of motor learning (e.g., progressing from grasping larger to smaller objects, reaching targets close to and then farther away within their workspace, mass to pincer grasp, navigating straight courses before progressing to slalom courses, increasing the number of obstacles in the path, etc.). Trainers followed a least-to-most prompting hierarchy, with the clear goal of promoting child independence during all training activities. The training was based on principles of motor learning, including promoting variable practice, discovery learning, and active problem solving, as well as providing multimodal feedback and timely reinforcement. 

#### 2.2.3. Testing Measures and Dependent Variables 

Our assessment battery included a combination of standardized tests, quantitative measures, and video-based measures of affected UE use during unimanual and bimanual tasks, both within and outside the training context. 

Standardized test: The Quality of Upper Extremity Skills Test (QUEST) [39] was administered at pretest and post-test to assess changes in affected UE range of motion and movement quality using 36 unimanual items administered on the affected and unaffected sides. The items fell within the 4 sub-domains of dissociated movements, grasps, protective extension, and weight bearing. This criterion-referenced test has been validated for children up to 12 years of age and shows excellent psychometric properties [40,41]. For this study, we calculated both the total and sub-domain scores. All scores on the QUEST are expressed as percentages (maximum value of 100), with higher scores indicative of better movement quality and range of motion. 

Quantitative measure of bilateral UE activity: To assess changes in the intensity and duration of affected and unaffected UE activity, as well as the level of asymmetry between both UEs, children were asked to wear activity monitors (wGT3X-BT, ActiGraph, Pensacola, FL, USA) on both wrists during the entire duration of the DJT sessions within the first and last weeks of the program. The ActiGraph activity monitors are small, light-weight sensors with 3-axis accelerometers (dynamic range ±8 gravitational units) that detect raw acceleration of UE movements in 3 directions with a sampling frequency of 30 Hz. Children wore the monitors once they were seated in the ride-on-toy, removing them at the end of each session when the toy was turned off; data collected were therefore only representative of UE movements. Children with data from at least 3 training sessions at each time-point (early and late weeks) were included in these analyses. Trainers also maintained wear time logs to document children’s adherence with wearing activity monitors during training sessions. 

Raw data from accelerometers were downloaded and processed through the ActiLife proprietary software v6.13.4, as well as a custom-developed code to calculate the extent of asymmetry in the duration and intensity of activity in the dominant versus non-dominant UEs [37,42]. The raw data were converted to activity counts (1 count = 0.001664 g, i.e., 0.0163072 m/s^2^) through ActiLife, while a composite metric summing activity counts in all 3 directions, called the vector magnitude (VM), was calculated ($VM=\sqrt{a_{x}^{2}+a_{y}^{2}+a_{z}^{2}})$**,** where, $a_{x}$, $a_{y}$, and $a_{z}$ are the accelerations or activity counts in the x-, y-, and z-directions, respectively) [37,42,43,44]. We also used the in-built Freedson children algorithm within the ActiLife software to classify average activity counts in the affected UE calculated over 60 s epochs across all training sessions [45], with the algorithm providing information on the percent duration of time spent by the affected UE in sedentary, light, moderate, and vigorous activity bouts specifically (sedentary: 0–149 counts/minute, light activity: 150–499 counts/minute, moderate activity: 500–3999 counts/minute, and vigorous activity: ≥4000 counts/min) [37,42,45]. 

For further analyses of raw data through our custom-designed software, data were downsampled to epoch lengths of 1 s. We calculated 2 metrics: 

(1) Use ratio indicative of relative duration of use (in hours) of non-dominant versus dominant UE, with values < 1 indicating greater use of the dominant UE, values > 1 indicating greater use of the non-dominant UE, and a value of 1 indicating equal use of dominant and non-dominant UEs.
$$Use\;ratio=\frac{hours\;of\;non-dominant\;or\;affected\;UE\;use}{hours\;of\;dominant\;or\;unaffected\;UE\;use}$$

(2) Magnitude ratio indicative of relative intensity of non-dominant versus dominant UE use per epoch, with negative values indicative of greater intensity of use of the dominant UE, positive values indicative of greater intensity of use of the non-dominant UE, and a value of 0 being indicative of a similar magnitude of use of the dominant and non-dominant UEs [46,47,48,49].
$$Magnitude\;ratio=\log\left(\frac{magnitude\;of\;non-dominant\;or\;affected\;UE\;activity}{magnitude\;of\;dominant\;or\;unaffected\;UE\;activity}\right)$$

We report on the mean and standard errors for the use ratio. For the magnitude ratio, given the skewness of these data, we report on the median value and interquartile ranges (IQR) [47]. 

We had the accelerometer data from both arms for 7 out of the 11 participating children and data from the affected arm for 8 out of 11 children. The missing data in the remaining children were either because they did not qualify the minimum wear time requirements for inclusion in these analyses (at least 3 training sessions in early and late weeks), technical failure of the watches, or due to research staff errors during data download.

Video-based measures of affected UE use: Video data from early and late training sessions were coded using Datavyu behavioral coding software v.1.3.8 to assess bimanual UE use, independent navigational abilities, and movement bouts. Raw coded data from Datavyu were exported into an excel format, while the excel sheets were post-processed using a custom-developed MATLAB program to calculate the dependent variables of interest. Each session was coded to calculate the percent duration of time during navigation (i.e., the ride-on toy is in motion) that the child was engaged in bimanual (both UEs used simultaneously to push/pull joysticks) versus unimanual (only unaffected or affected UE used) activity. We calculated a bimanual−unimanual activity ratio where higher values indicated greater bimanual compared to unimanual activity. Affected UE activity during navigation was further categorized as percent duration of navigation that the affected UE was engaged in independent (child pushed or pulled the joystick with their affected side independently without external trainer assistance), assisted (child required trainer assistance on the affected side to push or pull the joystick), or no activity (child’s affected UE was not pushing/pulling the joystick) bouts. We also coded the number of movement bouts initiated by the affected UE during navigation (a bout was defined as an effort exerted by the affected UE on the joystick, comprising one acceleration and one deceleration). We report on the rates of movement bouts during early and late training sessions. 

### 2.3. Statistical Analyses

Data were checked for assumptions of parametric statistics. We assessed the normality of our data by visualizing the data, calculating statistics on skewness and kurtosis, and using the Shapiro–Wilk test of normality (which is applicable for sample sizes < 50 participants) [50]. We used criteria provided by previous researchers that suggest that the data are considered to be normal if skewness is between −2 and +2 and kurtosis is between −4 to +4 [50,51,52]. Among our dependent variables, data on the QUEST, movement bouts, bimanual- unimanual ratio satisfied assumptions of normality. Two variables obtained from accelerometry data using the ActiLife software (specifically, percent time spent in sedentary bouts during early sessions and percent time spent in vigorous activity during late sessions) did not satisfy assumptions of normality. Similarly, for video-based measures of affected UE use, the percent time spent in independent and assisted activity did not satisfy assumptions of normality. For all the variables that violated assumptions of normality, we ran analyses using non-parametric Wilcoxon signed-rank tests. The results obtained from non-parametric tests were similar to the major trends observed using parametric statistics. Given the ease of interpretation of measures of central tendency and variability associated with parametric statistics, we report on data obtained from parametric statistics within the results Section 3.

We conducted a Pillai’s trace multivariate analysis of variance (MANOVA) to evaluate training-related changes in sub-domains of the QUEST. We used sub-domain (dissociated movements, grasps, weight bearing, and protective extension) and time-point (pretest and post-test) as the within-subjects factors. To analyze accelerometry measures obtained from the ActiLife software, we conducted a repeated-measures ANOVA with affected UE activity (percent time spent in sedentary, light, moderate, and vigorous activity) and time (early and late sessions) as the within-subjects measures. We used dependent *t* tests to analyze training-related changes in the use ratio and magnitude ratio. To analyze changes in independent navigational abilities, we conducted a repeated-measures ANOVA with affected UE activity (independent, assisted, and no activity) and time (early and late sessions) as the within-subjects factors. If the assumptions of sphericity were violated, as indicated by a significant Mauchly’s test, we used Greenhouse Geisser corrections. We also used dependent *t* tests to analyze changes in bimanual−unimanual activity ratio and rates of movement bouts from early to late sessions. For the MANOVA and ANOVA, if the analyses found a significant main effect and an interaction effect involving the same factors, we only evaluated the significant interaction effects. Please note that, based on our previous pilot work, we had identified specific planned comparisons of interest for each outcome measure a priori that we tested using dependent *t* tests. Statistical significance was set at a *p*-value of <0.05, and a statistical trend was reported for *p* values < 0.1. Effect sizes (ES) are reported for the MANOVA and ANOVA using partial eta-squared ${\;n}_{p}^{2}$ values (small effect = 0.01; medium effect = 0.06; and large effect = 0.14). For the planned comparisons using dependent *t* tests, we report effect sizes using Cohen’s *d* with Hedges correction [53]. For Cohen’s *d*, we report on ES estimates and 95% confidence intervals (CI) surrounding the point estimates. We classified Cohen’s *d* ES values according to Cohen’s conventions of small (0.2–0.49), medium (0.5–0.79), or large (0.8 and above) effects [54].

## 3. Results

### 3.1. Training-Related Changes in Standardized Test of Unimanual Motor Performance

The MANOVA for QUEST scores indicated a significant main effect of time (*F* (1, 10) = 37.90, *p* < 0.001, ${\;n}_{p}^{2}$ = 0.791) and a significant sub-domain × time (*F* (3, 8) = 13.48, *p* = 0.027, ${\;n}_{p}^{2}$ = 0.663) interaction. Analyses of the sub-domain × time interaction suggested that children demonstrated significant improvements on all four sub-domains of the QUEST from pretest to post-test (*t* values ranging from −2.51 to −3.27, *p* value range = 0.015 to 0.004). The ES for improvements ranged from medium to large (ES (95% CI)—Dissociated movements: 0.90 (0.20–1.58); Grasps: 0.73 (0.07–1.37); Weight Bearing: 0.82 (0.13–1.47); Protective extension: 0.95 (0.23–1.64)). Individual data suggest that all 11 children improved on total scores of the QUEST, with 6–9 children improving on each of the individual sub-domains of the QUEST (see Figure 3 and Table 1). 

### 3.2. Training-Related Changes in Accelerometry-Based Estimates of Bilateral UE Activity 

The repeated-measures ANOVA for affected UE activity measured using accelerometers suggested a significant main effect of activity type (*F* (1.316, 9.209) = 176.866, *p* < 0.001, ${\;n}_{p}^{2}$ = 0.962) and an activity type × session (*F* (3, 21) = 4.296, *p* = 0.016, ${\;n}_{p}^{2}$ = 0.380) interaction. Further analyses of the activity type × session interaction suggested that children demonstrated a significant increase in the percent duration of moderate activity (*t*(7) = −2.833, *p* = 0.025), with a concurrent trend for reduction in sedentary bouts (*t*(7) = 1.945, *p* = 0.09) of the affected UE from early to late sessions (see Figure 4 and Table 1 and Table 2). The improvement in moderate activity with the affected UE from early to late sessions was large in magnitude (ES (95%CI) = 0.947 (0.110–1.742)), with six out of eight children following the group trends. The reduction in sedentary bouts was medium-sized in magnitude (ES (95%CI = 0.650 (−0.103–1.367)), with six out of eight children followed the group trends and two children demonstrating a floor effect with no sedentary bouts during both early and late sessions (see Table 2).

The *t* tests to assess changes with training in the mean use ratio and median magnitude ratio did not reach statistical significance; however, it was encouraging to observe that, across both early and late training sessions, children with CP had use ratio values of ~1 (Mean (SE): Early = 1.01 (0.01), Late = 1.06 (0.07)), suggesting that the DJT sessions afforded equal duration of use of both UEs. Similarly, for the magnitude ratio, children showed a trend for improving the intensity of activity on their affected UE compared to their unaffected UE from early to late sessions (Median (IQR): Early = 0.29 (−0.34 to −0.17), Late = 0.00(−0.20 to 0.28)). 

### 3.3. Training-Related Changes in Video-Based Measures of Affected UE Use

The repeated-measures ANOVA for affected UE activity during navigation indicated a significant main effect of arm use (*F* (2, 20) = 33.17, *p* < 0.001, ${\;n}_{p}^{2}$ = 0.768) and an arm use × time interaction effect (*F* (2, 20) = 3.70, *p* = 0.043, ${\;n}_{p}^{2}$ = 0.270). Analyses of the arm use × time interaction suggested that, from early to late sessions, children showed a large increase in the percent duration of time of independent affected UE use during navigation (*t*(10) = −2.88, *p* = 0.016, ES (95% CI) = 0.84 (0.15–1.49); see Figure 5 and Table 1). Individual data suggest that 10 out of 11 children followed the group trend. 

A dependent *t* test suggested a significant large-sized increase from early to late DJT sessions in the bimanual−unimanual activity ratio (*t*(10) = −4.77, *p* < 0.001, ES (95% CI) = 1.38 (0.54–2.19); see Figure 6 and Table 1). Our data suggest that children increased the proportion of the DJT session spent in bimanual compared to unimanual activity from early to late training weeks, with all 11 children in the study following the group trends. There were no significant differences in the rates of movement bouts from early to late sessions (*t*(10) = 1.07, *p* = 0.31; see Table 1). 

## 4. Discussion

Our pilot study assessed the holistic effects of an innovative, technology-based training program that used dual-joystick-operated ride-on toys and was incorporated into an intensive, 3-week hybrid training protocol aimed at improving arm function in children with CP. Following the integrated training program, children improved their unimanual motor performance on a standardized motor test compared to baseline values. Between the early and late DJT sessions, children also increased the duration of moderate intensity activity with their affected arm and engaged in more equal, independent use of both UEs, as indicated by wrist accelerometry as well as video-based coding of training sessions. 

Our previous work has suggested that isolated motor training of the affected UE through goal-directed and purposeful navigational challenges using ride-on toys is feasible to implement, acceptable for stakeholders (children, clinicians, and caregivers), and can be seamlessly integrated into an intensive camp format for children with CP [36,55]. The current study results are encouraging and suggest that a DJT program may serve as an effective adjunct to conventional therapy in promoting bimanual motor skills. Such a DJT program offers several advantages: (1) ride-on toys afford opportunities for age-appropriate and inclusive play and may not be perceived as “therapy” by children with CP, (2) training activities are enjoyable and intrinsically rewarding to children, as indicated by high levels of session adherence and perceived enjoyment based on our past and current studies [36,37,55,56], and (3) use of ride-on toys may help incentivize children’s self-initiated active use of their UEs during functional, task-oriented games, which, when combined with conventional therapy, may achieve the optimal dosing required to harness neural plasticity to produce long-term changes in children’s UE motor function.

Our findings are in line with previous literature supporting the efficacy of task-oriented training based on motor learning principles for improving motor function in children with CP [57,58,59,60,61]. Task-oriented training emphasizes intensive, active practice of tasks that mimic functional daily activities rather than the mere repetition of so-called ‘normal’ movement patterns without purpose or meaning [62,63,64]. Such approaches provide children opportunities for agile problem solving, exploration and dynamic selection of movement patterns based on task and environmental constraints, as well as independent use of the affected UE for goal-directed meaningful activities within natural settings [65,66,67]. For example, Moon and colleagues compared the effects of a 4-week, 20-min/session, clinic-based task-oriented training program to usual care in children with hemiplegia, finding that the experimental group showed improvements in hand function measured using the box and blocks test, with no comparable improvements in the control group [58]. Along similar lines, our program was designed to promote gross and fine motor, multi-joint functional UE movement patterns involving push/pull, reach, throw/catch, grasp, release, and manipulation, which might have contributed to improvements in UE motor function.

The nature of the training context, i.e., the group camp format, may have also contributed to our findings. Although each child worked one-on-one with an instructor during the DJT sessions, the training was conducted in shared spaces within the camp premises that allowed each child to interact and receive encouragement/reinforcement from peer campers and adult camp staff. Moreover, older children in particular often tried to beat each other’s performance records during navigational games, which served as an added motivator for engagement in DJT activities. Past work from other groups has also suggested that group therapy seems to enhance child motivation and performance through peer interactions, positive feedback, competition, and behavioral modeling [57,68,69,70,71]. For example, group-based, task-oriented training provided for 1 h, 2 times/week for 8 weeks led to significant improvements in locomotor, manual dexterity, and social skills on standardized tests sustained at 16-weeks follow-up in children with spastic CP compared to a control group that received individual sessions of physical and occupational therapy [57]. Although effective in a group format, it remains to be tested if DJT sessions provided in an individual format can lead to improvements in UE outcomes in children with hemiplegia. 

Several research groups have explored the use of innovative technologies, such as virtual reality, augmented reality, exergames, and exoskeletons, as well as novel theme-based therapy models (e.g., magic-themed bimanual training camp, circus-themed hybrid mCIMT + BT camp) to promote engagement during intensive intervention protocols among children with CP [25,72,73,74,75,76,77,78,79,80,81,82]. For example, in a pilot study, Do and colleagues found that a 12-session virtual reality-based bilateral arm training program led to improvements in affected UE motor skill and use, as well as bimanual function in 3 children with hemiplegia, with gains sustained at 2 weeks after completion of the intervention [72]. In a different study, Gerber and colleagues evaluated the feasibility of a portable, computer-enhanced UE exergaming system called YouGrabber. Children wore neoprene gloves with mounted sensors as they engaged in computer games designed to train single and multi-joint UE movements. The data on UE position in space was collected through hand-mounted sensors that also provided haptic feedback to the child. Parents rated the system to be beneficial and feasible for home use in training affected UE motor functions [80]. Our findings add to this growing body of literature and suggest that joystick-operated ride-on toys may be used as playful therapy adjuncts to promote UE motor functions among young children with hemiplegia. 

It was encouraging that the DJT program along with other camp activities promoted more equal/symmetrical duration and intensity of bimanual UE use among children as indicated by the mean use ratio and median magnitude ratio values. From a neurological perspective, bimanual training leads to simultaneous activation of neural networks in both cerebral hemispheres and can therefore reduce the imbalance in hemispheric activation due to a lateralized lesion that is typically seen in children with hemiplegia [83]. In neurotypical individuals, corticospinal projections control contralateral movements, i.e., the right corticospinal tract controls left upper extremity movements and vice versa [84]. However, neural reorganization of the corticospinal tract may occur following lateralized perinatal injuries wherein some children may develop (a) ipsilateral corticospinal tract connections, meaning that the contralesional hemisphere may control both UEs, or (b) bilateral connections, where the affected UE may be controlled by both lesioned and contralesional hemispheres [85,86]. Nevertheless, work by Smorenburg and colleagues has demonstrated that bimanual training leads to improvements in unimanual and bimanual performance as well as functional outcomes in children with hemiplegia irrespective of corticospinal connectivity patterns [87]. Along these lines, bimanual training using dual-joystick ride-on toys may be used as an adjunct in rehabilitation practice to promote greater symmetry in duration and intensity of bilateral UE use among children with hemiplegia. 

### Limitations and Future Directions

Our pilot study used a single group, pre-post design without a control group, which significantly limits the conclusions we can draw related to the efficacy of the DJT paradigm. Instead, the study results reflect the combined effects of the existing hybrid programming at the camp and the supplemental DJT program. Our study is also limited by a small heterogenous sample recruited through convenience sampling. Our training, despite being intense (5 days/week), lasted for a relatively short duration. Moreover, we did not assess outcomes at follow-up to determine long-term carryover of improvements following completion of the program. We recommend that our study findings be replicated with larger sample sizes. In the future, we will conduct a larger two-group clinical trial to assess the isolated effects of a community-based DJT program compared to a dose-matched intervention based on conventional therapy on motor outcomes assessed immediately post-intervention as well as at 1- and 3 months follow-up. 

## 5. Conclusions

We conducted a pilot study to assess the effects of an innovative and child-friendly technology-based training program using dual-joystick-operated ride-on toys as an adjunct to intensive conventional UE rehabilitation for children with hemiplegia. Children received bimanual navigation training using ride-on toys as part of an intensive summer camp. Sessions were provided for 20–30 min/day, 5 days/week for 3 weeks. The holistic camp-based programming inclusive of DJT led to improvements in unimanual capacity on a standardized test at post-test compared to baseline. Moreover, from early to late training weeks, children increased independent use of their affected UE during sessions and engaged in more symmetrical activity (duration and intensity) with both UEs. Overall, our findings provide promising data to support the use of training programs using ride-on toys to incentivize affected UE active use and promote bimanual motor function among children with CP. 

## Figures and Tables

**Figure 1 bioengineering-11-00304-f001:**
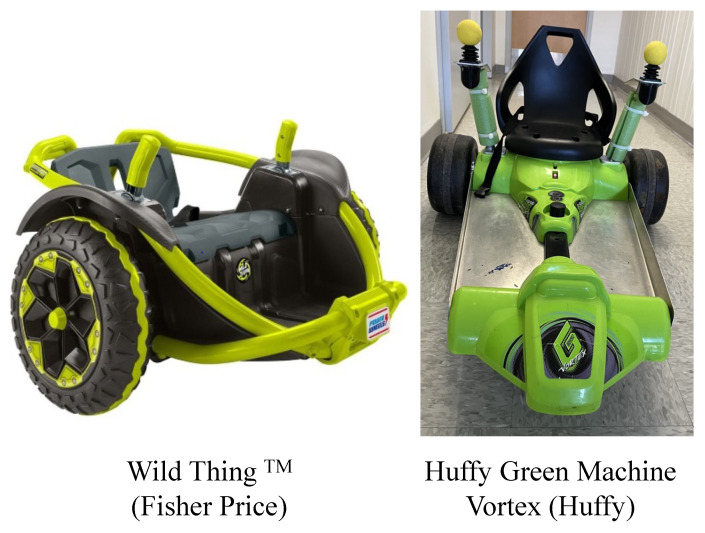
Modified, commercially available ride-on toys used in the study.

**Figure 2 bioengineering-11-00304-f002:**
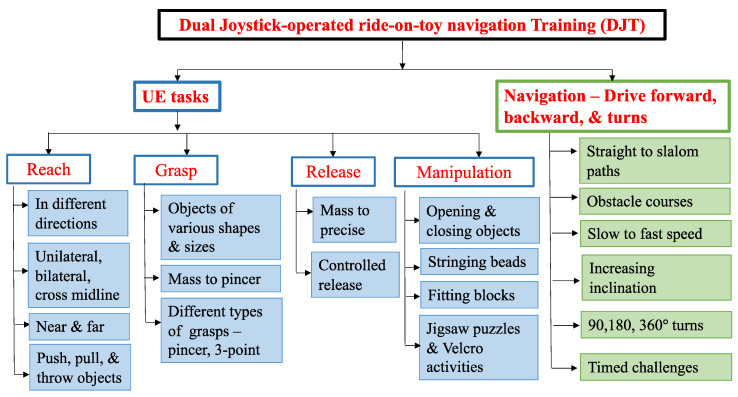
Framework for the manualized dual-joystick-operated ride-on-toy navigation training program.

**Figure 3 bioengineering-11-00304-f003:**
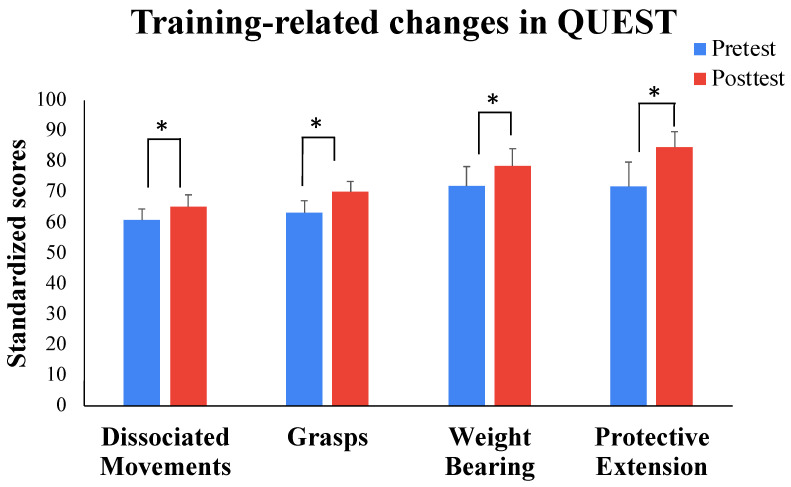
Training-related changes in scores on the standardized motor test (QUEST) assessed before and after the 3-week camp-based training program. * *p* ≤ 0.05.

**Figure 4 bioengineering-11-00304-f004:**
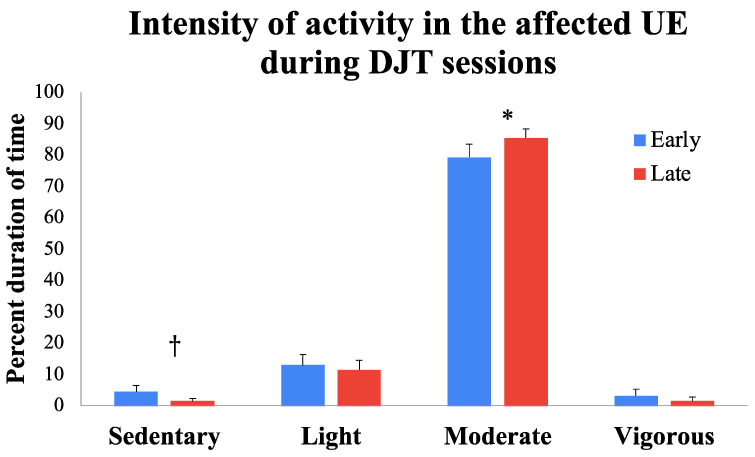
Training-related changes in the intensity of affected UE activity during early and late DJT sessions. * *p* ≤ 0.05, † *p* < 0.1.

**Figure 5 bioengineering-11-00304-f005:**
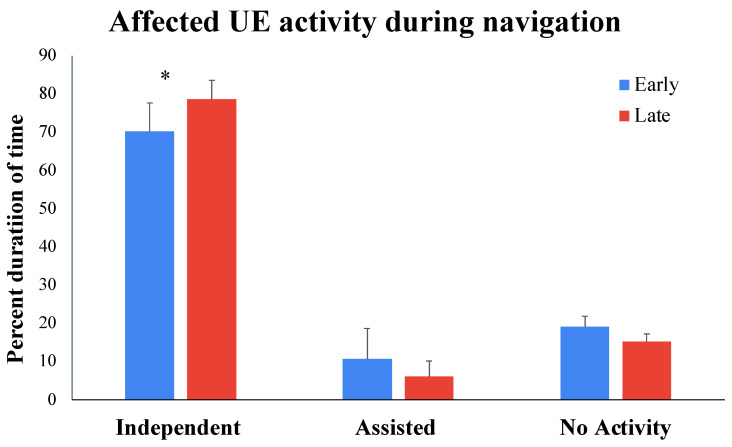
Training-related changes in the independent use of the affected UE for navigation from early to late sessions. * *p* ≤ 0.05.

**Figure 6 bioengineering-11-00304-f006:**
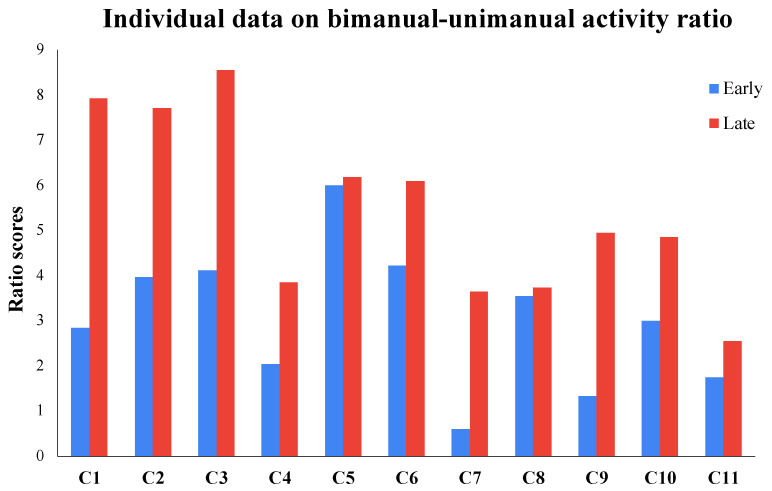
Training-related changes in relative proportion of bimanual to unimanual activity during navigation in early and late sessions; Note: “C” stands for “Child” and higher activity ratio values indicate greater bimanual compared to unimanual activity.

**Table 1 bioengineering-11-00304-t001:** Summary of training-related changes in measured outcome measures in the study. * *p* ≤ 0.05, † p<0.1.

Outcome Measures	Pretest/Early Session (Mean (SE)) or Median (IQR)	Post-test/Late Session (Mean (SE)) or Median (IQR)
Quality of Upper Extremity Skills Test (QUEST) (pretest to post-test)		
Dissociated movements	60.90 (3.47)	65.20 (3.88) *
Grasps	63.30 (3.82)	70.03 (3.29) *
Weight bearing	72 (6.38)	78.55 (5.58) *
Protective extension	71.72 (7.96)	84.60 (5.06) *
Accelerometry-based measures of affected UE activity (early to late sessions)		
% sedentary time	4.53 (2.08)	1.52 (0.75) †
% light activity	12.93 (3.38)	11.47 (2.96)
% moderate activity	79.3 (4.21)	85.5 (2.86) *
% vigorous activity	3.25 (1.99)	1.47 (1.3)
Mean use ratio	1.01 (0.01)	1.06 (0.07)
Median magnitude ratio (IQR)	0.29 (−0.34 to −0.17)	0.00 (−0.20 to 0.28)
Video-based measures of affected UE activity during navigation (early to late sessions)		
% independent use	70.14 (7.50)	78.64 (4.91) *
% assisted use	10.77 (7.79)	6.07 (4.10)
% no activity	19.09 (2.78)	15.30 (1.99)
Rates/minute of bouts	39.64 (4.49)	37.53 (5.45)

**Table 2 bioengineering-11-00304-t002:** Individual data on training-related changes in intensity of activity in the affected UE measured during DJT sessions using wrist-worn accelerometers.

Intensity of Affected UE Activity	% Time Spent Sedentary		% Time in Light Intensity Activity		% Time in Moderate Intensity Activity		% Time in Vigorous Intensity Activity	
Child	Early	Late	Early	Late	Early	Late	Early	Late
1	1.14	0	98.86	97.73	0	2.273	0	0
2	0	0	78	80.18	19.95	19.82	2.05	0
3	0	1.25	83.69	91.84	12.82	6.912	3.49	0
4	0	0	76.55	86.36	16.48	9.468	6.97	4.17
5	0.69	0	70.56	74.39	24.49	22.18	4.26	3.43
6	0	0	60.04	75.95	22.04	19.52	17.9	4.53
7	14.6	0	76.22	88.43	7.678	11.57	1.52	0
8	9.61	10.5	90.39	89.49	0	0	0	0

## Data Availability

Data presented in this manuscript are available upon request from the first author.

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
