# Peer review of "A Training Program Using Modified Joystick-Operated Ride-on Toys to Complement Conventional Upper Extremity Rehabilitation in Children with Cerebral Palsy: Results from a Pilot Study"

_bioengineering, 2024, doi:10.3390/bioengineering11040304_

Round 1

Reviewer 1 Report

Comments and Suggestions for Authors

In this paper the authors present a feasability/pilot study using commercialy available gaming technology to perfrom rehabilitation on children with cerebral palsy.

The paper is well writen but I am not sure that it i best fitted for Bioengineering journal. The auhtors should justify why they decide to submit this sutdy in this particular journal.

Overall comments:

Please present the results in a Table with the results of the statistical analysis to complement the graphics.

If you want to publish in Bioengineering you should put much more emphasis on the data processing part since this is – IMO – the only engineering component on this study.

Statistics: How did the authors assessed the normality of the data? With such a small sample size I am not sure of the normality of the data and I am sure that trying to find interaction in such sample size is not possible (or at least highly underpowered). It’s a pilot study so why not focus on simple pre-post test comparison and discuss the feasibility of such intervention?

Did you adjust for multiple testing?

3.1. Missing data – should be presented in the methods and in the description of the sample.

Minor

Figure legend should always be below the figures

Lines 195: Bilateral

Author Response

Please see the attachment below

Reviewer 2 Report

Comments and Suggestions for Authors

This paper presents the results of a pilot study using dual-joystick-operated ride-on toys to improve arm function in children with cerebral palsy. The topic is important and relevant, and the paper is written clearly in general. Some major and minor comments follow.

Major comments

1. The conclusions of this study are weak since there was no control group. The data were collected before and after a 3-week camp and all participants in the study were using the dual-joystick-operated ride-on toys. The study design does not permit making any conclusions about whether the toys were beneficial, detrimental, or had no effect. For example, the conclusion "...the DJT program promoted more equal/symmetrical duration and intensity..." (line 432) is not supported by the results of this study.

2. The first author has published several papers on this topic. Three papers are cited on line 85 ([35]-[37]) but no commentary is provided. It would be helpful to briefly summarize the findings of the studies that the author has already published on this topic and highlight the novelty of the present paper.

3. On lines 341-343, a decrease in the "assisted" and "no activity" categories (Fig. 5) is noted despite the fact that there is no statistical significance. This should never be done. The purpose of the statistical test is to determine whether the numbers are different. If there is no statistically significant difference, it means that repeating the study could very well result in the opposite relationship between the numbers. Noting "trends" because one number is greater than the other is potentially misleading; this is the very reason we use statistical tests.

Minor comments

4. The caption for Fig. 1 should appear immediately below the figure.

5. At lines 217-219, 4000 counts/minute is not included in any category.

6. In Fig. 4(a), the meaning of the asterisk and dagger should be provided in the caption.

7. Fig. 4(b) does not appear to add as much value as the space it occupies. Perhaps this panel could be replaced with a table showing the percent duration of time for each participant and each intensity level.

8. References contain many errors. For example, important words in journal titles must be capitalized. Some journal names are abbreviated while others are written in full. Several citations are missing the journal name (e.g., 62, 67, 69-71, 74-76, 78, 79, 82, 83).

Comments on the Quality of English Language

Typos should be corrected. For example, "a paper" -> "a piece of paper" (line 44), "every day" -> "everyday" (line 51), "carryover" -> "carry over" (line 70), remove black rectangles (lines 101, 119, 120 and 125), "bislateral" -> "bilateral" (line 195), "down sampled" -> "downsampled" (line 221), "atleast" -> "at least" (line 277).

Author Response

Please see attachment below

Round 2

Reviewer 2 Report

Comments and Suggestions for Authors

The authors have adequately addressed the concerns I raised in my first review (Reviewer 2). The modified title, explanation of previous work at lines 85-93 and removal of conclusions that are not supported by the results of this study make the message of this paper clearer. Some minor issues the authors may wish to address before proceeding with publication:

1. Much of the paper uses full justification but some parts do not. A consistent format is recommended.

2. Figure 2 should not extend into the margin.

3. Lines 224-225: the "a" variables are not boldface in the equation but they are boldface when defined on line 225. The same notation should be used if these are the same variables.

4. Lines 239 and 246: the hyphens appear to be subtraction symbols. This text should be entered in text mode.

5. Line 246: the parentheses should be taller.

6. Line 254: delete the underscore at the end of the line.

7. Line 258: "excel" should be "Excel".

8. Line 279: "between -4 to +4" should be "between -4 and +4".

9. In Figure 3, the vertical lines could be centered above the respective bars.

10. Lines 339, 341, 347, 350 and possibly others: closing parentheses are missing. There should be as many ")" as there are "(".

11. References should be formatted consistently. The following references have paper titles that are capitalized inconsistently compared to the other references: 20, 22, 36, 37, 42, 59, 76, 77 and 79.

12. Lines 511-514 and 666: delete extra periods (search for "..").

Comments on the Quality of English Language

Please see corrections above.